# Shedding Light on Viral Shedding: Novel Insights into Nuclear Assembly, Cytoplasmic Transformation and Extracellular Vesicle Release of the BK Virus

**DOI:** 10.3390/ijms25169130

**Published:** 2024-08-22

**Authors:** Daniela Gerges, Karim Abd El-Ghany, Zsofia Hevesi, Monika Aiad, Haris Omic, Clemens Baumgartner, Wolfgang Winnicki, Michael Eder, Alice Schmidt, Farsad Eskandary, Ludwig Wagner

**Affiliations:** 1Division of Nephrology and Dialysis, Department of Internal Medicine III, Medical University of Vienna, 1090 Vienna, Austria; daniela.gerges@meduniwien.ac.at (D.G.); karim_abdel-ghany@outlook.com (K.A.E.-G.); n01618704@students.meduniwien.ac.at (M.A.); haris.omic@meduniwien.ac.at (H.O.); clemens.baumgartner@meduniwien.ac.at (C.B.); wolfgang.winnicki@meduniwien.ac.at (W.W.); michael.eder@meduniwien.ac.at (M.E.); alice.schmidt@meduniwien.ac.at (A.S.); ludwig.wagner@meduniwien.ac.at (L.W.); 2Center for Brain Research, Medical University of Vienna, 1090 Vienna, Austria; zsofia.hevesi@meduniwien.ac.at; 3Division of Endocrinology, Department of Internal Medicine III, Medical University of Vienna, 1090 Vienna, Austria

**Keywords:** BK polyomavirus (BKPyV), tubular epithelial cell, BKPyV infection, capsid protein VP1, kidney transplant (KTX)

## Abstract

Despite the high prevalence of BK polyomavirus (BKPyV) and the associated risk for BKPyV-associated nephropathy (BKPyVAN) in kidney transplant (KTX) recipients, many details on viral processes such as replication, maturation, assembly and virion release from host cells have not been fully elucidated. VP1 is a polyomavirus-specific protein that is expressed in the late phase of its replicative cycle with important functions in virion assembly and infectious particle release. This study investigated the localization and time-dependent changes in the distribution of VP1-positive viral particles and their association within the spectrum of differing cell morphologies that are observed in the urine of KTX patients upon active BKPyV infection. We found highly differing recognition patterns of two anti-VP1 antibodies with respect to intracellular and extracellular VP1 localization, pointing towards independent binding sites that were seemingly associated with differing stages of virion maturation. Cells originating from single clones were stably cultured out of the urine sediment of KTX recipients with suspected BKPyVAN. The cell morphology, polyploidy, virus replication and protein production were investigated by confocal microscopy using both a monoclonal (mAb 4942) and a polyclonal rabbit anti-VP1-specific antibody (RantiVP1 Ab). Immunoblotting was performed to investigate changes in the VP1 protein. Both antibodies visualized VP1 and the mAb 4942 recognized VP1 in cytoplasmic vesicles exhibiting idiomorphic sizes when released from the cells. In contrast, the polyclonal antibody detected VP1 within the nucleus and in cytoplasm in colocalization with the endoplasmic reticulum marker CNX. At the nuclear rim, VP1 was recognized by both antibodies. Immunoblotting revealed two smaller versions of VP1 in urinary decoy cell extracts, potentially from different translation start sites as evaluated by in silico analysis. Oxford Nanopore sequencing showed integration of BKPyV DNA in chromosomes 3, 4 and 7 in one of the five tested primary cell lines which produced high viral copies throughout four passages before transcending into senescence. The different staining with two VP1-specific antibodies emphasizes the modification of VP1 during the process of virus maturation and cellular exit. The integration of BKPyV into the human genome leads to high virus production; however, this alone does not transform the cell line into a permanently cycling and indefinitely replicating one.

## 1. Introduction

BK polyomavirus (BKPyV) is a highly prevalent virus with oligo- or asymptomatic primary infection that occurs most often during childhood or adolescence, leaving individuals with lifelong protective immunity [1,2]. However, due to its permanent residence in the urogenital tract after primary infection, and certain virus-specific oncogenic potential, uncontrolled reactivation due to immunosuppression may cause BKPyV-associated nephropathy (BKPyVAN) and rarely even urogenital malignancy in recipients of kidney transplants (KTX) [3,4]. Although common, the exact mechanisms of infection, replication and potential integration into the host genome remain incompletely understood. Inter-individual variation with respect to the extent of viral shedding during BKPyV infection and/or reactivation may be related to such genetic rearrangement within the viral genome but also upon site-specific integration into the host genome, temporarily enhancing viral fitness until viral clearance by immune reconstitution is re-established [5,6]. To shed light on possible clinical implications that arise from such modifications and to better define potential therapeutic strategies, a more deepened understanding of viral integration and shedding is key [7].

BKPyV primarily replicates in the epithelial cells of the urogenital tract. After infecting the cell, the BKPyV genome enters the host-cell nucleus, where early gene expression and replication of the mini-chromosome occur, followed by late gene expression. The virus utilizes the host’s replisome to replicate its double-stranded DNA, and virion assembly subsequently takes place in the nucleus [8,9]. While integration in the host’s genome has been reported, especially in patients with BKPyV-positive cancers, integration is not common, and it remains unclear which factors and alteration in gene expression are contributing [10,11,12,13].

The mode of viral shedding that has been most frequently described in the literature is cell lysis, where exuberant viral replication leads to cell rupture. It commonly occurs in the context of BKPyVAN in immunocompromised KTX patients. Infected tubular epithelial cells with high viral load undergo a lytic process and exfoliate from the tubular basement membrane and appear in the urine as so-called decoy cells [14,15,16]. However, recent reports indicate that non-lytic viral shedding might be possible [17]. BKPyV-infected tubular epithelial cells appear to release infectious particles via extracellular vesicles (EVs), which can traffic tens of viral particles [18]. Other members of the *Polyomaviridiae* family include the JC polyomavirus (JCPyV), simian virus 40 (SV40) and the Merkel cell polyomavirus. Such EVs have also been described for the genetically related JC polyomavirus (JCPyV), which utilizes EVs to target host cells and spread infection [19]. Regarding SV40, it is clinical practice that a SV40-specific antibody against the large T antigen is utilized to stain kidney biopsy samples for the immunohistological diagnosis of BKPyVAN. In SV40, various viral proteins such as VP4 appear to regulate exocytosis [20,21]. VP4 was initially discovered as a VP2-gene-derived product translated from an alternate translation start site downstream from that of the genuine VP2 [20].

Release enveloped in host membranes is thought to enhance virion stability and immune evasion [22], ensuring an uncontained spread of infection, unnoticed by the host’s immune response [23,24].

In this work, we investigated the replication and means of viral spreading of BKPyV within tubular cells, specifically highlighting possible modifications of the virus and its proteins as it traverses the cell. Our primary aim was to establish patient-specific cell culture models derived from urinary cell sediment that show active BKPyV replication, which might provide insight into replicative patterns based on individual genetic alterations of the virus that might also include potential chromosomal insertion sites. To detect differences in viral packaging, nuclear egress patterns and intracellular accumulation of virus-like particles, we included the use of two specific antibodies targeting the main viral protein VP1 that were characterized in their distinct staining behavior throughout intra- and extracellular virus production and release.

## 2. Results

### 2.1. Patient Demographics

Twenty-one patients of European descent were screened for this study, of which stable cell cultures from urinary sediment were established in nine patients (Table 1 and Table 2). Two of them showed active ongoing BKPyV replication until senescence (Table 3). Median age was 58 years (27–78 years), and two (22.2%) were female. Prior KTX had occurred in two (22.2%) patients, and all others were recipients of a first KTX. Eight patients were on tacrolimus bid (median tacrolimus target-trough level at the time of sampling: 5.2 ng/mL) and one patient (#7) received 4-weekly belatacept infusions (5 mg/kg at the time of sampling). Seven patients received a reduced dose of mycophenolate mofetil (MMF, 500 mg, bid) due to ongoing BKPyV replication. In one case, MMF was paused due to BKPyVAN, while another patient (#8) received azathioprine (150 mg daily). At the time of sampling, plasma BKPyV load and urine BKPyV load ranged between 1.4 × 10^2^ and 3.6 × 10^4^ copies and 7.0 × 10^4^ and 3.0 × 10^10^, respectively. KTX biopsy results were available in all patients, and BKPyVAN diagnosis was established according to the Banff Working Group Classification of Definitive BKPyVAN: five patients were classified as PVN class 1, one patient as PVN class 2 and three patients as PVN class 3 [25]. Immunohistochemistry for SV40 was performed on six of nine tissue samples. Only one biopsy sample stained positive for SV40 (Table 1 and Table 2).

### 2.2. BKPyV Replicates in Non-Immortalized Tubular Cell Lines

In order to procure cells capable of BKPyV replication and to facilitate further investigation into its biology, urinary sediment cells obtained from 21 patients diagnosed with BKPyVAN were cultured. Among these, nine cell lines were successfully established, exhibiting variations in cell morphology and growth kinetics (Appendix A), yet all reaching confluence. Although the replicative potential differed among cell lines, the percentage of tetraploid cells (M-Phase) in each population ranged between 17 and 25% (Figure 1A–C). As detailed in Table 3, identification of the clonally expanding cells of renal tubular epithelial origin was analyzed by positive immunofluorescence for TMEM27 as well as negative labelling for the urinary bladder-specific cytokeratin 13 (KRT13). The cells continued to expand and grow over a period of 15 days to 2 months even after repeated cryopreservation. They were split between 3 and 15 times, and stable BKPyV replication was established in a total in six out of nine cell lines (Table 3). Two of them showed high BKPyV secretion (#6 and #7) from clonal expansion until they reached senescence, and four of them were re-infected.

Following the establishment of nine stably cultured urinary cell isolates from nine different individuals that exhibited active in vitro replication of BKPyV, it was of further interest to characterize BKPyV’s intracellular maturation, assembly and shedding utilizing an anti-VP1-specific antibody for further visualization.

### 2.3. Vesicles Enable BKPyV Shedding from Infected Tubular Cells

Our next goal was to describe the mode of infection and viral shedding of BKPyV within the tubular cell cultures in order to differentiate between low and high replicators. Therefore, the supernatant of the four BKPyV producing cultured patient-derived decoy cells was utilized to infect those cell lines, which were lacking the formation and secretion of mature BKPyV (#4, #5, #8, #9). As mentioned above, two different antibodies recognizing the VP1 antigen were applied to visualize the viral distribution and shedding in these cell culture specimens.

In order to characterize the localization of VP1 within our BKPyV-infected tubular cell culture model, we applied two different anti-VP1 antibodies with variations in their epitope specificities. We had observed that our polyclonal antibody might recognize also earlier translational products of VP1 involved in different stages of viral particle maturation, whereas the monoclonal antibody (mAb 4942) would be more specific for a mature conformational VP1 epitope. Staining with the polyclonal RantiVP1 Ab (green) revealed localization of nuclear virus assembly factories and the perinuclear region, suggesting a possible concentration in the endoplasmic reticulum (ER) (Figure 2).

In contrast, the staining pattern of mAb 4942 predominantly exhibited a granular pattern within the cytoplasm and immediate nuclear periphery (Figure 3). It also showed high specificity in the detection of decoy cells [26]. Additionally, it revealed extracellular virus-loaded vesicles following cell rupture, which were observed in typical idiomorphic sizes and BKPyV-loaded vesicles immobilized on cytopreparations (Figure 4). The staining pattern also varied significantly with the cell cycle stage of the individual cells (Figure 5).

Staining of both antibodies appeared to overlap at the nuclear rim, as a site of viral accumulation before egress through the nuclear membrane as shown in Figure 3 and Figure 4 in yellow. The staining pattern of mAb 4942 may suggest that this antibody recognizes the VP1 protein in the later stages of viral production and egress.

To further clarify the different recognition patterns, RantiVP1 was tested for colocalization with a marker protein of the ER, CNX. As indicated by the shaded staining pattern, VP1 was recognized by RantiVP1 at the ER production stage and clearly colocalized with the ER marker protein CNX, as shown in Figure 6.

To visualize the cytoplasmic membrane of VP1-positive cell lines, dual-color staining for AQP1 and VP1 using mAb 4942 was performed. In Figure 7, cells with VP1-positive vesicles appeared to have an intact cytoplasmic membrane. The cell with an enlarged nucleus (white arrow) showed reduced VP1 staining.

As the distribution of the BKPyV genotype in our patient cohort may reflect that of Europeans, the anti-BKPyV VP1 reactivity of the rabbit polyclonal antibody against both genotype I and genotype IV had to be confirmed and established; it is shown in Figure 8.

### 2.4. Decoy Cells Exhibit Truncated Forms of the VP1-Gene Product

Recent findings by Henriksen et al. indicated that SV40 and BKPyV produce a truncated VP2/3, denoted VP4 [27], and this led us to perform tests to investigate whether a similar truncated form of VP1 might exist. To investigate possible changes in the VP1 protein, immunoblotting was conducted on both decoy cells and mature virions isolated from patients’ urine.

Immunoblotting of decoy cells with the polyclonal RantiVP1 antibody confirmed VP1 at its established 41 kDa weight and also identified two additional reactive bands at 36–37 and 17 kDa. The highest expression levels were observed in extracts from decoy cells. Thus, it appears that two additional *VP1*-gene derived products (VP1a and VP1b at 36–7 kDa and 17 kDa, respectively) were expressed in decoy cells (Figure 8).

In contrast, immunoblotting of mature BKPyV virions isolated from patients’ urine, with high viral load (>10^7^ copies/mL), showed only a single VP1 band at 41 kDa. This indicates that the VP1 variants, VP1a and VP1b, identified in patient decoy cells are likely not present in mature secreted virions (Figure 9).

### 2.5. Identification of Possible Translation Products from VP1 mRNAs by In Silico Analysis

The VP1 cDNA sequence was analyzed using an initiation codon identification server. A total of six potential translation products were identified. As indicated in Figure 10, the product with the highest probability represented the genuine 362-amino-acid-long VP1 protein. However, among the other potential products, one consisted of 307 amino acids with a calculated molecular weight of 34.23 kDa, and another consisted of 173 amino acids with a calculated molecular weight of 18 kDa (Figure 10). These two variants could correspond to the experimentally identified products VP1a and VP1b, respectively, as observed in the immunoblots. The size variants observed in the immunoblot, when compared to the in silico analysis, can only be explained by additional post-translational modifications. These modifications may include proteolytic cleavage or glycosylation and phosphorylation. Other variants identified by in silico analysis when including BKPyV genotype I and IV were not present in immunoblotting experiments.

### 2.6. BKPyV DNA Integrates into the Host Cell

As a final step, we thought it would be of interest to investigate whether integration into the host genome of BKPyV had occurred in our cell culture models. BKPyV DNA has been reported to occasionally integrate into the host genome, affecting all chromosomes except the Y chromosome, primarily in patients with BKPyV-related tumors [12,13]. We investigated whether such integration was also present in our study’s cell lines. Of particular interest was whether differences in gene integration existed between cells that produce low numbers of viral copies and those that secreted high levels of mature virus particles. Therefore, we chose to compare the two cell lines with permanent BKPyV-replication and production (cell lines #6 and #7) to three of the primarily non-infected cell lines (cell lines #2, #4 and #5 from an early non-infected passage) through ONT long read sequencing.

No integration was detected in patient cell lines #2, #4 and #5. Sample #7 had insufficient sequencing coverage to confirm integration. However, sample #6 exhibited significant viral genome accumulation on chromosomes 3, 4 and 7 (Table 4, Figure 11). At these sites the BKPyV genome could be identified and presented in relation to the genetic surroundings using the IGV server (Figure 11, Table 4). On chromosome 3, the BKPyV genome was found to be integrated into the TAFA1 gene, which plays a role in chemotaxis. On chromosome 4, the BKPyV genome was integrated into a noncoding region between the Pyroglutamylated RFamide peptide receptor (QRFPR) and Annexin A5 (ANXA5) genes. On chromosome 7, the BKPyV genome was found integrated into an intron of the inner mitochondrial membrane peptidase subunit 2 (2IMMP2L) gene.

## 3. Discussion

Infections with BKPyV are highly prevalent in the general population, affecting over 90% of individuals, typically without clinical manifestation. However, in the immunocompromised, especially KTX patients, BKPyV may lead to organ failure and graft loss of the transplanted organ as well as cause tumors. To date, the precise mechanisms of BKPyV infection and propagation, including its cell-to-cell transmission and potential disparities in viral shedding between immunocompetent and immunosuppressed hosts, remain inadequately characterized.

In this study, we successfully established a BKPyV infection model in newly cultured tubular progenitor cell lines. These cells were directly isolated from the source tissue via urine samples and importantly were neither immortalized nor transformed in any way. This natural propagation provides a more accurate representation of BKPyV’s behavior in native cellular environments, as shown in detail earlier [28]. Furthermore, using two VP1-specific antibodies with different recognition patterns, we could discern viral changes during its maturation and to our best knowledge visualize BKPyV shedding via EVs, confirming data of earlier authors [18]. Additionally, in one case we proved BKPyV integration into three chromosomes of the host genome.

In the generation of the cell lines, we demonstrated that BKPyV from culture supernatant of urinary decoy cells can be used to establish infection, while this could not be achieved using isolated virions from urine. In doing so, we established a BKPyV infection model in non-immortalized tubular progenitor primary cells lines from KTX patients, which has some novelty. Experiments on infectivity and virus tropism were conducted by other authors using primary human epithelial cells, epithelial cells, endothelial cells and fibroblasts derived from the respiratory and urinary tracts. These experiments were performed to assess the susceptibility and specific cellular targets of the virus across different tissue types [10]. Further experiments were performed using immortalized tubular cell lines expressing human telomerase reverse transcriptase [29]. Our tissue culture model might originate from the changes in tubular cell dynamics and presence of tubular progenitor cells in urine due to BKPyV infection. As kidney injury diminishes and viral load decreases, the renal tubules heal and undergo replication, causing replicating tubular epithelial cells to appear in urine [30,31,32]. Our research shows these progenitor cells expanded effectively from a clonal phase and continued BKPyV replication and secretion.

Following virus replication and translation, BKPyV undergoes post-translational modifications in the cytoplasm, which are critical for virion assembly and future infectivity [22]. While cell lysis is one known mechanism of BKPyV spread, envelopment of the virus into the lipid bilayer of the host’s cell membrane represents a different mode of virus egress, which bears advantages for the virus [18,24,33]: (1) Protection: the lipid envelope provides an additional layer of protection for the viral genome, keeping it from degradation by host nucleases and the hostile environment [34]. (2) Cell entry: Enveloped virions can enter host cells by different mechanisms than non-enveloped virions. The lipid envelope can facilitate fusion with cell membranes or promote endocytosis, potentially increasing the efficiency of viral entry [35]. (3) Evasion of the immune system: The envelope consists of the host’s cell membrane; therefore, viruses traveling within EVs evade recognition and neutralization by the host’s immune system [24]. Vice versa, the elimination of viruses in vesicles out of a cell might serve as a defense strategy for individual cells; however, it comes at the cost of a considerable risk to the entire organism due to the reasons mentioned above. Up to now, the precise mechanisms governing EV formation at the virus–cell interaction level remain incompletely elucidated [36]. In our study, mAb 4942 successfully detected mature virions packaged in vesicles.

Pertaining to our potential identification of two additional modified *VP1*-gene products, similar observations have been made by other authors with respect to the *VP2*-gene product. For VP2, a truncated protein named VP4 has been described, which arises due to alternate translational start sites of VP2, and it appears that VP4 might be involved in progeny release [27]. In this context, several different polyomavirus proteins have been described as viroporins such as agnoprotein, VP2, VP3 and VP4 [37,38,39]. These proteins form small channels in host cell membranes and can alter cellular processes, including calcium signaling and cell cycle regulation, which are crucial for viral replication and release. VP4 has been shown to be of importance for viral entry into the host cell and to trigger the timely lytic release of BKPyV’s and SV40′s progeny [20,27]. In our study, the origin of the two additionally identified *VP1*-gene products—whether through alternative translation variants or protease cleavage—and their potential beneficial function to BKPyV must be left unanswered. Further studies will have to investigate if VP1a/VP1b exhibit functions similar to VP4.

Regarding virus genome integration, earlier studies confirm fragmentary integration of BKPyV at multiple sites into the host genome allowing the virus to persist in host cells unrecognized by the immune system and undergo transcription and translation of viral proteins [12,13]. This phenomenon has been observed in the monkey kidney-derived COS cell line, which is capable of transcribing and translating the large T antigen protein of SV40 but does not produce capsid proteins or mature virions [26]. Our data confirmed the integration of BKPyV DNA into three different chromosomes in one cell line, but no driver insertion was found which has been linked to carcinoma formation in the past [12]. Certain genes could be disrupted in this cell line, but this is not necessary for the maintenance of a cell line culture.

A significant observation that is important for clinical considerations is the verified integration of the BKPyV genome into three different human chromosomes in one of the cell lines. Despite this integration, there was no activation of pertinent cell cycling pathways, and the cell line underwent growth arrest after passage 4. This suggests that BKPyV integration alone does not directly induce malignant transformation. It is likely that a specific genomic environment is required to facilitate a driver-type integration that leads to oncogenesis. These findings highlight the complexity of BKPyV-associated cellular transformation and underscore the need for further research to elucidate the precise conditions and mechanisms that enable viral integration to contribute to cancer development.

In conclusion, the maturation process of the BKPyV under these culture conditions is orchestrated towards envelopment into vesicles for egress, a mechanism that enhances infectivity and allows evasion of the immune system. Two additional *VP1*-gene-derived products exist, which are only found within the host cell during virus replication of so-called decoy cells. Utilizing freshly established tubular progenitor cells derived from patients with manifest BKPyVAN provides a valuable platform for studying BKPyV biology. Furthermore, these cells offer the opportunity to identify genomic integration of the virus, facilitating deeper investigations into the mechanisms of BKPyV infection and the search for inhibitory compounds.

## 4. Materials and Methods

### 4.1. Cell Culture and Cell Lines

Kidney transplant patients with a positive serum BKPyV copy number under observation at the outpatient clinic of the Division of Nephrology and Dialysis at the Medical University of Vienna starting May 2023 till end of January 2024 were sequentially included. Individuals providing urine volumes with less than 10^5^ cells in total were excluded from this study. Out of 982 screened KTX patients for BKPyV-uria, 75 were eligible for this study. However, 54 were excluded as they showed less than 10^5^ cells in the provided urine sample.

Twenty-one patients were enrolled after providing informed consent. The immunosuppression was reduced due to BKPyVAN. From nine of them—classified as PVN Class 1 (n = 5), PVN Class 2 (n = 1) and PVN Class 3 (n = 3)—clonal cell outgrowth could be achieved [25]. KTX biopsy results were available for all patients. Clinical routine charts provided laboratory parameters, urine and serum BKPyV copy numbers and details about the immunosuppressive regimen.

The morning urine (80 mL) of these patients was centrifuged at 2000× *g* rounds per minute (rpm) for 10 min. The resulting cell pellet was resuspended in RPMI 1640 culture medium containing 10% fetal bovine serum, penicillin and streptomycin. The cells were seeded in Cellstar 60 × 15 mm tissue-culture-treated dishes (Greiner bio-one) and incubated at 37 °C, 5% CO_2_ and 95% humidity.

The medium was changed after 3 days to proximal tubular cell culture media (REBM/REGMTM) obtained from Lonza (Lonza Group AG, Basel, Switzerland). Normocin (InvivoGen, San Diego, CA, USA; Cat. # ant-nr-05) was added as an antibiotic. After 6–7 days, small clones comprising 3–9 cells in adherent growth mode were observed. These clones increased in size due to cell proliferation. As indicated in Appendix A, when expansive colonies with confluence and rapid growth were reached, a first cell passage was performed using trypsin-EDTA 1x in calcium/magnesium-free PBS (Biowest, Nuaillé, France, L0940-100) without phenol red. Similar methods for culturing urinary renal epithelial cells have been described by other authors [30].

This study was approved by the Ethics Committee of the Medical University of Vienna under the number 1065/2021.

### 4.2. BKPyV Copy Number Evaluation

BKPyV copy number was evaluated as detailed by Pajenda et al. [40]. In brief, to evaluate BKPyV copy number, cDNA or DNA was mixed with Universal Master Mix (Applied Biosystems, Waltham, MA, USA) and the BKPyV probe set (Pa03453401_s1, Applied Biosystems, Waltham, MA, USA). The PCR reaction was performed on a QuantStudio 6 Flex qPCR machine (Applied Biosystems, Waltham, MA, USA). A sample was considered positive if it reached the amplification threshold by cycle 38.

A tenfold serial dilution of a plasmid containing the BKPyV-specific qPCR target sequence (Pa03453401_s1) was used as a standard series. Copy numbers of positive samples were calculated according to this standard series. As an alternative and for clinical samples, the test kit from GeneProof^®^, Brno, Czech Republic (GeneProof BK Virus (BKV) PCR Kit—IVDR) was utilized.

### 4.3. Obtaining BK Polyoma Virus by Ultracentrifugation

Urine samples of patients with BKPyVAN were pelleted using an Optima Max-XP tabletop ultracentrifuge (Beckman Coulter, Brea, CA, USA). Therefore, one milliliter of each sample was applied to an open-top thick wall polycarbonate centrifuge tube, and symmetrically aligned tubes were centrifuged at 100,000 rpm for 2:30 h in a TLA-120.2 fixed-angle rotor. Times for acceleration and deceleration between zero and 5000 rpm were set to 2:30 min and 6:00 min, respectively.

### 4.4. Cell Line Infecton Using BKPyV-Containing Medium from Decoy Cell Culture

To prepare the virus-containing medium, decoy cells obtained from urine of patients were incubated at a concentration of 5 × 10^4^ cells/mL in culture medium (REBM/REGMTM) for 3 days. The culture fluid was then centrifuged for 10 min at 3000× *g* to pellet cellular fragments. The clarified supernatant was tested for BKPyV copy number (10^8^ copies/mL) and stored in aliquots at 4 °C for immediate use. For longer storage periods, the aliquots were kept at −80 °C. This tissue culture medium containing high copy numbers of BKPyV was used for infecting renal epithelial cell lines.

Semiconfluent (50–60%) renal epithelial cell lines from patients #4, #5, #8 and #9 were exposed to virus-containing medium (2 × 10^7^ copies/mL) for 24 h. Following this incubation period, the cell cultures were passaged, and the culture medium was replaced with fresh culture medium.

### 4.5. Immunoreagent

A polyclonal antibody against BKPyV-VP1 was generated by immunizing a female rabbit (*Oryctolagus cuniculus*) with recombinant His-tagged VP1 protein expressed in *Escherichia coli* (*E. coli*) and purified using Ni-NTA agarose (Qiagen, Hilden, Germany), as described in Pajenda et al. [40]. The amino acid sequence of the protein and its coding gene sequence are provided in Appendix A in the attached file. Serum was collected after five injections of the purified VP1 protein. Anti-VP1 IgG was subsequently purified from the serum using either a protein G column or ammonium sulfate precipitation. The polyclonal antibody was used in 0.8–1 µg/mL. In addition, a second mouse monoclonal VP1-specific antibody (mAb, 4942) was purchased from ThermoFisher (Invitrogen, Carlsbad, CA, USA, Catalog # MA5-33242) and used in 1–2 µg/mL. This mAb was generated by the use of BKPyV-VP1 as immunogen and has been validated and tested for staining BKPyV-infected decoy cells by immunofluorescence in an earlier work [26]. The collectrin (TMEM27)-specific antibody PA5-88721 (Invitrogen) was used in a dilution ratio of 1:100. The Cytokeratin 13-specific mAb CL488-66684 (Proteintech, Rosemont, IL, USA) was diluted at a ratio of 1:400 in PBS. The mouse monoclonal antibody specific to calnexin (CNX) was purchased (Catalog # MA3-027) from ThermoFisher, Waltham, MA, USA. The anti-aquaporin 1 (AQP1) rabbit polyclonal Ab (AB2219, diluted) was purchased from Millipore, Burlington, MA, USA.

### 4.6. Immunofluorescence by Confocal Microscopy

Confluent cells were liberated from the tissue culture dish using trypsin as described above. Following inactivation of trypsin with culture medium, a cell suspension of 10^5^ cells/mL was generated. Out of this, 60 µL was applied to the funnel of the cytocentrifuge, which was run at 1200 rpm for 3 min. The resultant cytopreparation was air-dried for 1 h followed by fixation in acetone for 4 min. A water repellent cycle was drawn around the cell-containing area, and the primary antibody was applied. The mAb 4942 anti-human BK-VP1 was diluted at 1:30, and the rabbit anti-human BK-VP1 was diluted at 1/1000 and incubated for 2 h at room temperature or at 4 °C overnights. Following a washing step in PBS for 10 min under constant stirring, the second antibody was applied to either goat anti-mouse Alexa Fluor 594 (diluted at 1:700 in PBS) or goat anti-rabbit Alexa Fluor 488 (diluted at 1:700 in PBS). DAPI for nuclear staining was contained in the secondary antibodies. After a second washing step in PBS, slides were mounted with Vectashield mounting medium for immunofluorescence (Vecotor Laboratories, Burlingham, CA, USA) and covered with a microscope glass coverslip for confocal microscopy. A 3D confocal digital assembly was achieved by performing Z-stack recording comprising 40 individual scans, with a scan density of 0.2 µm, capturing imaging data from the top to the bottom of the nuclear staining. The recorded images were subsequently processed into the 3D format using ZEN blue (Version 3.4 for Windows 10, Carl Zeiss Microscopy, LLC, White Plains, NY, USA). Samples from BKPyV-negative patients served as negative controls.

### 4.7. Immunoblotting

The cell pellet (2000 rpm) was resuspended in PBS and pelleted again already in the sample grinding tubes containing the DNA-binding/grinding resin. Following a brief centrifugation and removal of the supernatant, the extraction solution (1x sample buffer containing DTT and protease inhibitors, cOmplete Tablets, Mini, REF 04 693 124 001) was added, and cells were broken up using the pestle (Cytiva, 80-6483-37). The resultant cell lysate was spun at 12,000× *g* for 7 min, and the precleared supernatant was loaded onto a BioRad PAGE mini gel (Hercules, CA, USA) for protein separation. Following protein separation, the gel was transferred onto nitrocellulose by semidry blotting. The blotted membrane was blocked in blocking buffer (Pierce^TM^ Protein-Free T20 Blocking Buffer, Appleton, WI, USA, 37571) for 20 min, and rabbit anti-human BK-VP1 was added (diluted at 1:1500) and incubated over night at 4 °C. After washing the blot with TPBS twice for 10 min, the goat anti-rabbit HRP conjugate (Dako P0448, Agilent, Sana Clara, CA, USA, diluted at 1:10,000 in PBS substituted with 10% blocking buffer) was incubated for 60 min at room temperature under constant shaking. Finally, the blot was developed for specific antibody binding after washing again with TPBS twice for 10 min using chemiluminescence reagent (Merck KGaA, Darmstadt, Germany) and recorded by the lumi-imaging device Fusion FX Vilber Lourmat (Vilber, Eberhardzell, Germany).

### 4.8. Virus DNA Isolation

For virus DNA isolation from culture supernatant, the InnuPREP Sewage Water DNA/RNA Kit (Innuscreen Gmbh, Berlin, Germany) was used. In brief, 1 mL of supernatant was transferred to a 2.5 mL tube, followed by the addition of 100 μL of VCR-1 and 100 μL of VCR-2 reagent. The mixture was then incubated at room temperature for 10 min. Subsequently, centrifugation at 13,000 rpm was performed for 10 min. The pellet was then washed with 1 mL of RNase-free water and then dissolved in 150 μL of Lysis Solution RL, which was followed by the addition of 10 μL of Proteinase K. The tube was placed in a thermomixer and shaken continuously for 15 min at 60 °C. Subsequently, 135 μL of binding solution SBS and 10 μL of MAG suspension M were added. After thorough mixing and incubation for 5 min at room temperature with constant rotation, the suspension underwent two washes with 300 μL of wash solution HS and two washes with 300 μL of wash solution LS. Following the final washing step, the solution was removed as completely as possible, while ensuring the retention of the beads. The beads were then dried at 50 °C for 15 min. Finally, the DNA was eluted with 15 μL of RNase-free water.

### 4.9. DNA Isolation from Human Cell Lines and Sequencing by Oxford Nanopore Technology

Cell line cells were suspended in PBS and DNA isolation was performed using the QIAamp^®^ DNA Blood Mini Kit 50 protocol (Qiagen, Hilden, Germany), following the guidelines outlined in the test manual. The column-purified DNA was subsequently submitted for Oxford Nanopore Technology (ONT) sequencing at the Vienna Biocenter core facility. After undergoing quality checks, the obtained reads were utilized for bioinformatics analysis, specifically focusing on the integration of the virus into the human host genome.

### 4.10. Bioinformatics Identification of BK Polyoma Virus Integration

Basecalling of raw data (pod5-format) from the promethION sequencing device was performed with ONT’s basecaller Dorado [https://github.com/nanoporetech/dorado]. Samples were aligned against the viral reference sequence available at https://www.ncbi.nlm.nih.gov/nuccore/OQ230873.1, using Minimap2 v2.24 [https://github.com/lh3/minimap2], (accessed on 20 February 2024), (minimap2 -t 16 -ax mapont $viralReference $reads). Subsequently, only reads that aligned to the viral reference sequence with a mapping quality of at least 60 were retained after filtering (“filteredViralReads”). Filtering and format conversion were performed using samtools v1.18 [https://samtools.github.io/], (accessed on 20 February 2024). In order to identify putative insertion sites, the viral reads were aligned against the canonical chromosomes of the human genome available at https://www.ncbi.nlm.nih.gov/datasets/genome/GCF_000001405.40/, using the splice-aware alignment mode of Minimap2 (accessed on 20 February 2024), with the following command: minimap2 -t 16 -ax splice $humanReference $filteredViralReads. The IGV hg18 database for localizing the genetic environment of integration was accessed on 20 February 2024 (https://igv.org/app/). 

### 4.11. In Silico Analysis of Potential Translation Initiation Sites on the VP1 mRNA

The freely available calculator from the Helix Research Institute (https://atgpr.dbcls.jp) was used to calculate the translation initiation probability. The 1089 nucleotide sequence encoding the VP1 genotype I and IV cDNA was entered into the server to identify potential open reading frames with possible translation start sites.

## Figures and Tables

**Figure 1 ijms-25-09130-f001:**
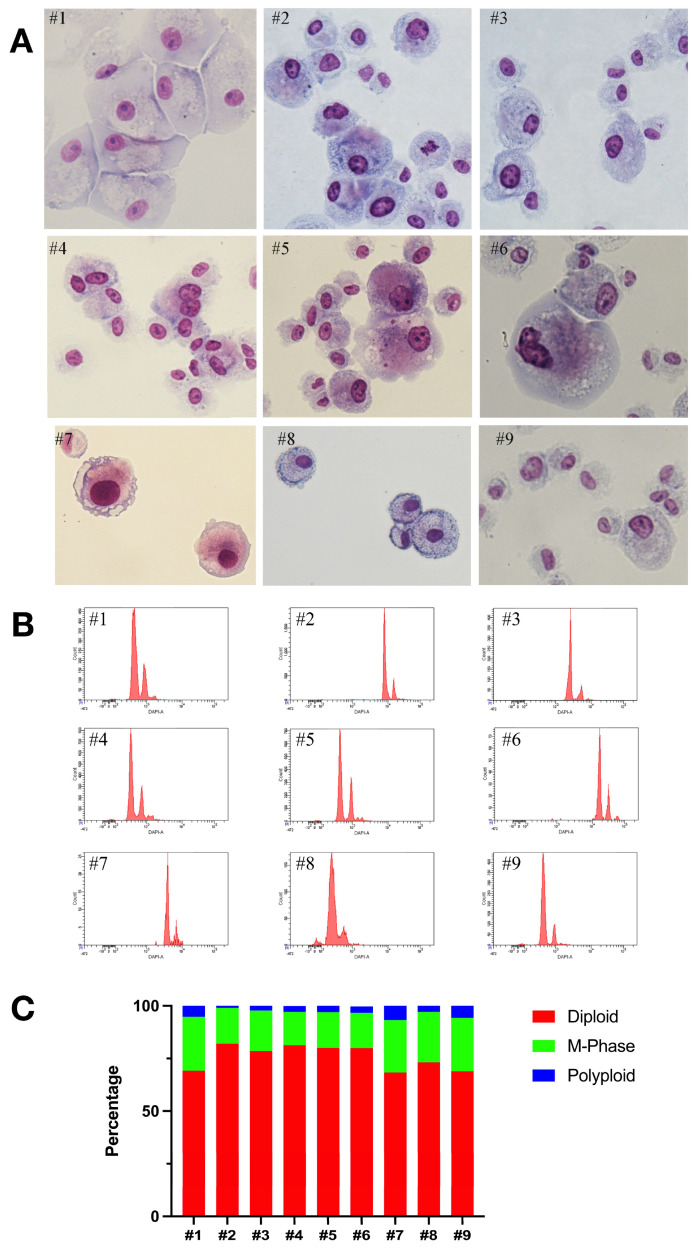
Hematoxylin/Eosin staining, FACS analysis and distribution of DNA content of primary cell lines. (**A**) HE staining exhibited diploid cells, cells with enlarged nucleus (DNA synthesis phase) and polyploidy as depicted in patients #4 and #6, (orig. magnification 400×). (**B**) Flow cytometry (FACS) analysis of cellular DNA content was performed on all nine established primary tubular cell lines. The cells were permeabilized and stained with DAPI prior to analysis. (**C**) Distribution of DNA content in diploid, tetraploid (M-phase), and polyploid cells, expressed as a percentage of the total cell population, was determined by DAPI staining of permeabilized cell line cells. The data are presented for each of the nine patients.

**Figure 2 ijms-25-09130-f002:**
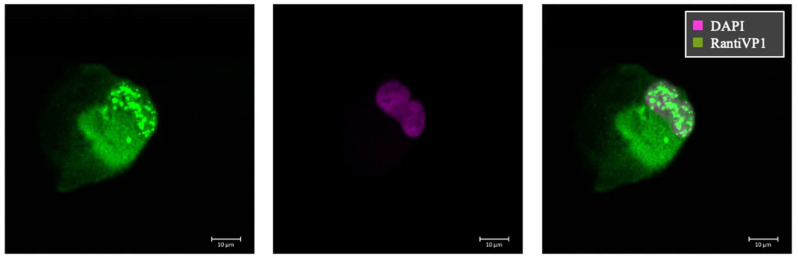
Confocal immunofluorescence for VP1 protein. Immunofluorescence for VP1 in patient #1 derived cell line using RantiVP1 (green) and DAPI (purple). Nuclear punctate staining was observed, representing “nuclear virus assembly factories”, and shadowed staining in the cytoplasm, indicating VP1 translation and production after passage #1.

**Figure 3 ijms-25-09130-f003:**
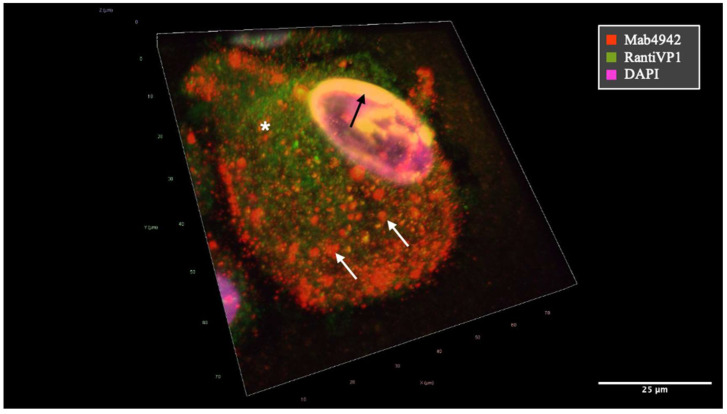
Merged image showing colocalization of VP1 as detected by mAb 4942 (red) and the RantiVP1 (green) resulting in yellow coloration. Confocal fluorescence microscopy revealed the process of virus particle formation using two distinct VP1-specific antibodies (mAb 4942 = red, RantiVP1 = green). BKPyV-VP1 encapsulated within small vesicles traversing the cytoplasm, ultimately accumulating at the cellular periphery was mainly stained by mAb 4942 (red) (annotated by white arrows). Conversely, the VP1-specific rabbit antibody (RantiVP1) highlighted the VP1 protein localization in the cytosol and in proximity to the nucleus (white asterisk) at a different stage of VP1 production and maturation. Virus formation within nuclear regions is depicted as accumulation at the nuclear rim by both antibodies resulting in yellow staining (annotated by black arrow).

**Figure 4 ijms-25-09130-f004:**
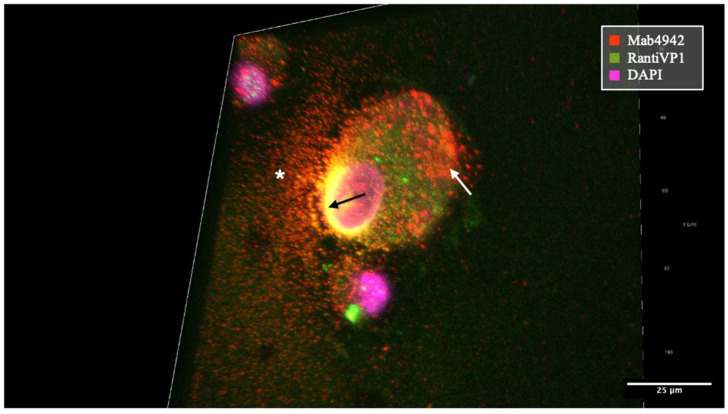
VP1 staining encapsulated within small vesicles released from a virus-producing cell of patient #7’s primary cell line. VP1-containing vesicles as stained by mAb 4942 were distributed throughout the cytoplasm (white arrow). At the nuclear rim, the BKPyV-VP1 was prominently recognized by both antibodies mAb 4942 and RantiVP1 resulting in a high degree of overlap and yellow coloration (indicated by small, long-tailed black arrow). Extracellular VP1-loaded vesicles stained with mAb 4942 after cell rupture, in idiomorphic sizes, immobilized on the cytopreparation, marked by a white asterisk.

**Figure 5 ijms-25-09130-f005:**
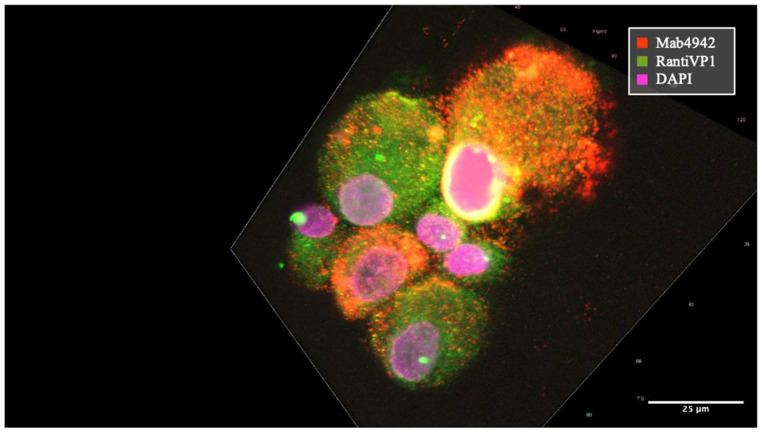
BKPyV-infected cells in different stages of the cell cycle. Two cells exhibited the presence of mature enveloped virus particles dispersed throughout the cytoplasm, predominantly of uniform size (mAb 4942 = red). The cell positioned at the top of the image displayed virus accumulation at the nuclear rim, where it was co-recognized by both antibodies (yellow). In contrast, in four other cells, VP1 staining, predominantly recognized by RantiVP1 antibodies (green) was observed, with a minor presence of virus particles stained by mAb 4942.

**Figure 6 ijms-25-09130-f006:**
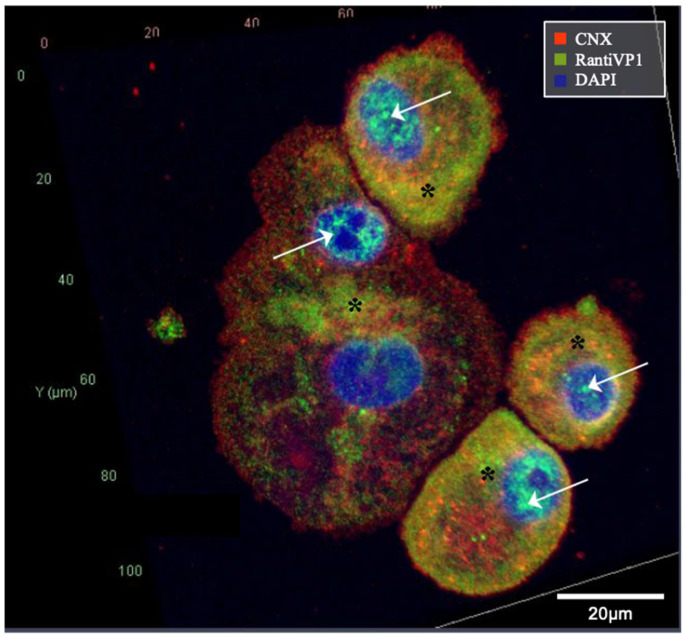
Confocal BKPyV-VP1 staining and colocalization with ER marker CNX. The VP1 detected by the RantiVP1 (green) colocalizes (yellow, indicated by *) with the ER marker CNX (red) in younger cells of the cell cycle. VP1 staining at the nuclear virus assembly factories is present in the smaller cells and in one of the nuclei of the large tetraploid cells (indicated by arrow).

**Figure 7 ijms-25-09130-f007:**
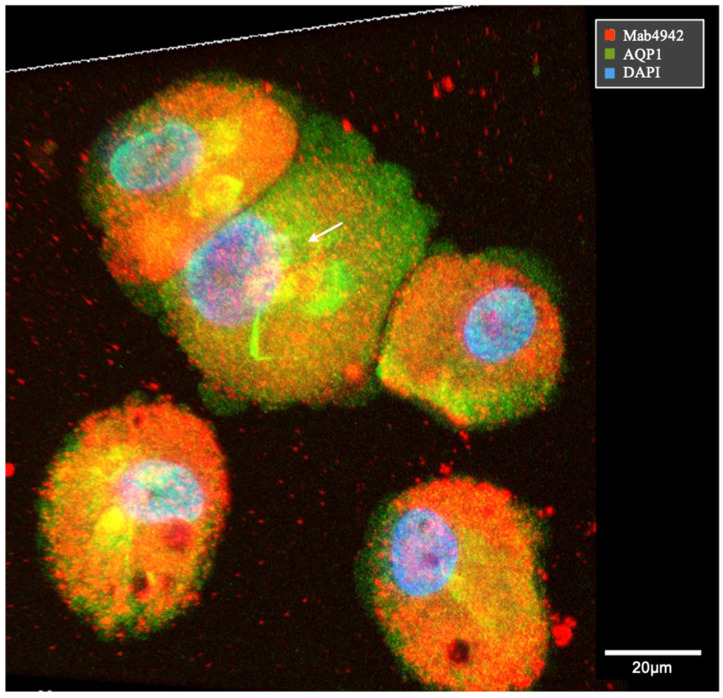
Confocal VP1 staining and visualization of the plasma membrane. VP1 positivity was detected by mAb 4942 (red), and the cytoplasmic membrane is visualized by AQP1 (green). The cell with an enlarged nucleus, marked by a white arrow, showed reduced VP1 staining.

**Figure 8 ijms-25-09130-f008:**
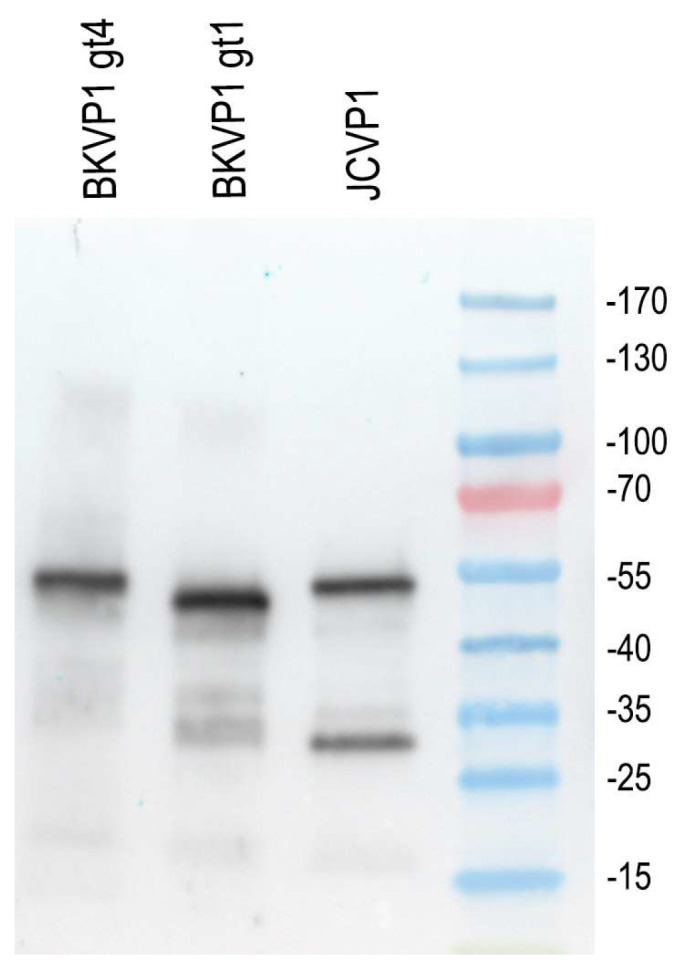
Comparative analysis between BKPyV VP1 genotype I and genotype IV using recombinant protein expressed in *E. coli* using RantiVP1. JCPyV VP1 protein recognition is depicted in the lane next to the molecular weight marker protein. VP1 proteins produced in *E. coli* from the JCPyV and BKPyV, followed by purification utilizing the His tag and Ni NTA resin. The slightly different sizes of the recombinant variants are attributed to the additional amino acids representing the Tag depending on the expression plasmid used (see Appendix A).

**Figure 9 ijms-25-09130-f009:**
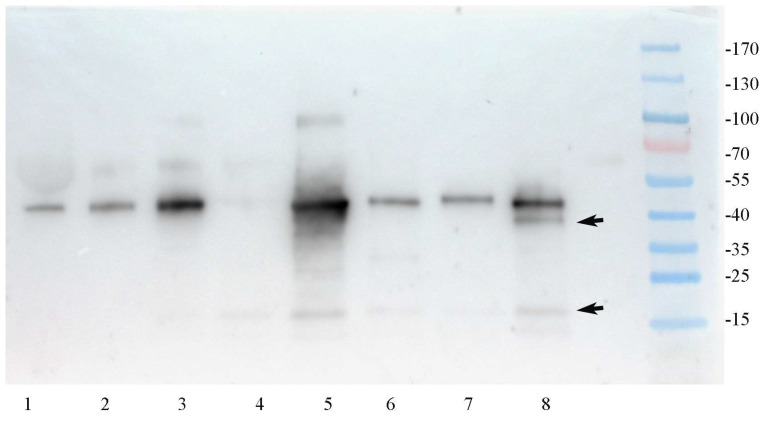
SDS-Page and immunoblot analysis of BKPyV virions as pelleted from urine by ultracentrifugation and protein extracts from urine sediment with decoy cells showed genuine VP1 and VP1 variants. Lanes 1–3 display urinary virion extracts pelleted by ultracentrifugation from patients with a urine viral load of 10^7^–10^9^ copies/mL. No additional VP1-specific smaller bands were detectable in any of the urine virion isolates. From Lanes 4 to 8, extracts of 2 × 10^4^ urinary cells were loaded into each lane. Lane 4 shows cell extract from a patient recovered from disease but still the presence of faint VP1 bands. Lane 5 represents urine sediment obtained from a patient at the time of diagnosis, exhibiting over 50% decoy cells, with two variant bands detected: one at approximately 100 kDa representing aggregates and another at 16–17 kDa. Conversely, Lanes 6 and 7 depict patients in the recovery phase, showcasing less than 10% decoy cells among urine sediment cells. Lane 8 illustrates the VP1 banding pattern of a patient with 30% decoy cells in the sediment, wherein the 16–17 kDa band resembles that of Lane 5, albeit with an additional band approximately 2–3 kDa lower in size than the genuine VP1 protein.

**Figure 10 ijms-25-09130-f010:**
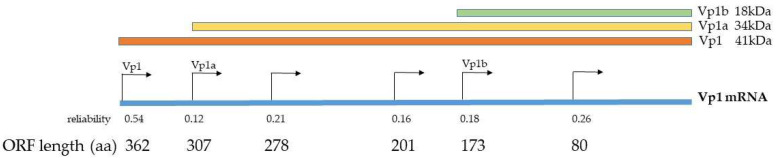
In silico analysis for potential translation products originating from the BKPyV-VP1 mRNA revealed several possible products. The highest reliability score of 0.54 was assigned to the genuine VP1 protein, which is 362 amino acids in length. Additionally, two other potential products, VP1a and VP1b, with lengths of 307 and 173 amino acids, respectively, were also identified. These reached reliability scores of 0.12 and 0.18.

**Figure 11 ijms-25-09130-f011:**
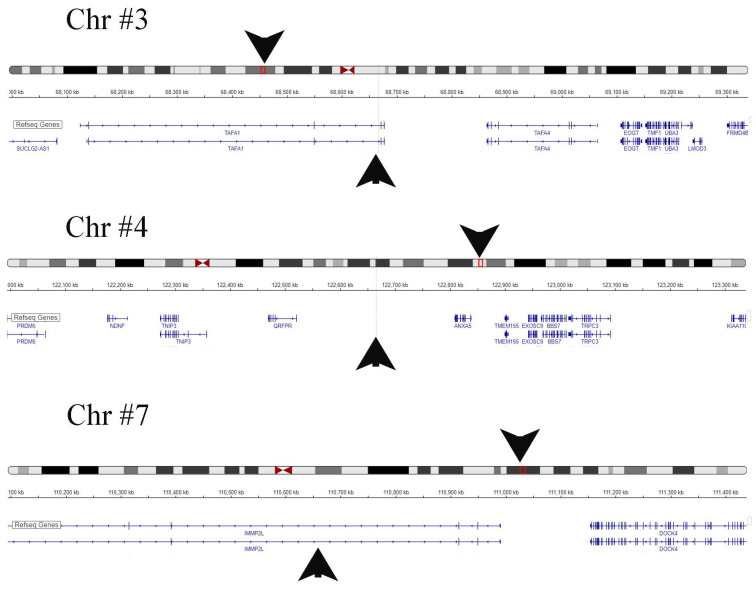
Schematic representation of BKPyV integration sites on chromosomes 3, 4 and 7, identified by ONT in patient #6. By passage three, the patient’s cell line secreted virions at a logarithmic count of 10^10^/mL. Post-passage four, a decline in cell growth rate occurred. Chromosomal integration sites are marked by red box, with black arrows indicating their positions relative to adjacent genes, which are listed in Table 4. The illustration was created using the publicly accessible IGV hg18 database.

**Table 1 ijms-25-09130-t001:** Patient demographics.

Characteristics	All Patients (n = 9)
Age, y	58 (36–58)
Male	7 (77.8)
First KTX	7 (77.8)
Second KTX	2 (22.2)
Duration after last KTX, mo	10 (9–15)
Plasma BKPyV load, copies per mL	11,566 (11,459–11,674)
Urine BKPyV load, copies per mL	7.02 × 10^9^ (5.65 × 10^9^–8.38 × 10^9^)
**Immunosuppression**	
TAC	8 (88.9)
MMF	7 (77.8)
Azathioprine	1 (11.1)
Belatacept	1 (11.1)
Prednisolone	9 (100)
**BKPyVAN** [25]	
PVN class 1	5 (55.6)
PVN class 2	1 (11.1)
PVN class 3	3 (33.3)
**SV40**	
Positive	1 (11.1)
Negative	5 (55.6)
Not performed	3 (33.3)

Values are median with 1st and 3rd quartile in brackets (Q1–Q3). Immunosuppression is given at time of sampling. BKPyV = BK polyomavirus; BKPyVAN = BK polyomavirus associated nephropathy; KTX = kidney transplantation; mo = months; mL = milliliters; SV40 = simian vacuolating virus 40; y = years.

**Table 2 ijms-25-09130-t002:** Patient demographics. Age in years, numbers (#) of KTX; duration after last KTX in months (duration); immunosuppressive regimen with tacrolimus and mycophenolate mofetil; percentage of decoy cells in urine (HE), urinary BKPyV load in copies per mL, plasma BKPyV load in copies per mL, SV40 positivity in KTX biopsy and PVN class according to Nickeleit V et al. specified by KTX biopsy [25].

ID	Age (y)	Sex	# of KTX	Duration (mo)	TAC	MMF	HE (%)	uBKPyV (Copies/mL)	pBKPyV (Copies/mL)	SV40	PVN Class
1	52	m	1	13	Y	Y	93	8.7 × 10^5^	1.4 × 10^2^	-	1
2	55	f	2	75	Y	Y	0	5.8 × 10^6^	2.0 × 10^3^	-	3
3	27	m	1	15	Y	Y	30	1.7 × 10^7^	1.5 × 10^4^	NA	1
4	72	m	1	10	Y	Y	10	4.3 × 10^9^	3.6 × 10^4^	-	3
5	60	f	2	24	Y	Y	0	5.2 × 10^8^	8.3 × 10^3^	NA	2
6	78	m	1	9	Y	Y	50	3.0 × 10^10^	2.0 × 10^4^	+	3
7	56	m	1	8	N	N	0	7.0 × 10^4^	2.2 × 10^2^	NA	1
8	63	m	1	10	Y	N	30	1.8 × 10^9^	4.5 × 10^3^	-	1
9	58	m	1	8	Y	Y	30	1.9 × 10^9^	1.6 × 10^4^	-	1

f = female; HE = hematoxylin eosin; ID = identification; m = male; mL = milliliters; MMF = mycophenolate mofetil; mo = months; NA = not available; pBKPyV = plasma BKPyV load; PVN = polyomavirus nephropathy; SV40 = simian vacuolating virus 40; TAC = tacrolimus; uBKPyV = urinary BKPyV load; y = year, # = number, + = positive, - = negative.

**Table 3 ijms-25-09130-t003:** Viral infection model showing integration into host DNA, VP1-positivity with both polyclonal Ranti-VP1 and the mAb 4942, cells type identification to either urinary bladder epithelia (KRT13) or the kidney (TMEM27), the infective potential of cell lines (infectivity) and number of passages after which cells exhibited growth arrest (senescence).

ID	Integration	RantiVP1	mAb 4942	KRT13	TMEM27	Infectivity	Senescence
1	NP	+	-	-	+	NP	3
2	NI	+	-	-	+/-	NP	3
3	NI	+	-	-	+	NP	3
4	NI	+	-	-	+/-	+	5
5	NI	-	-	-	+	++	15
6	Chr 3,4,7	+	++	-	+	Infected	5
7	NA	+	++	-	+/-	Infected	4
8	NP	-	-	-	+	+	14
9	NP	+/-	-	-	+/-	+	5

Chr = chromosome; integration = viral integration into host DNA; RantiVP1 = polyclonal rabbit antibody; mAB 4942 = commercial VP1-specific antibody; NA = not applicable due to insufficient sequencing coverage; NI = no integration detected; NP = not performed; KRT13 = cytokeratin 13; TMEM27 = collectrin; + = infective, ++ = highly infective; infected = infected itself with BKPyV.

**Table 4 ijms-25-09130-t004:** Exact BKPyV integration into chromosomes 3, 4 and 7. Location of integration (location), genetic relation, disruption of host genes and supporting reads identified by ONT in patient #6.

Chromosome	Location	Genetic Relation	Host Genes	Supporting Reads
3	68666797	Intragenic	TAFA1	2123
4	122666278	Intergenic	QRFPR-ANXA5	171
7	110766605	Intragenic	IMMP2L	58

ANXA5 = Annexin A5; IMMP2L = inner mitochondrial membrane peptidase subunit 2; QRFPR = Pyroglutamylated RFamide peptide receptor; TAFA1 = Chemokine like family member 1

## Data Availability

The original contributions presented in the study are included in the article/Appendix A, further inquiries can be directed to the corresponding author/s.

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
