# Peer review of "Shedding Light on Viral Shedding: Novel Insights into Nuclear Assembly, Cytoplasmic Transformation and Extracellular Vesicle Release of the BK Virus"

_ijms, 2024, doi:10.3390/ijms25169130_

Round 1

Reviewer 1 Report

Comments and Suggestions for Authors

Review of ijms-3091366

Shedding light on viral shedding: Novel Insights into Nuclear Assembly, Cytoplasmic Transformation and Extracellular Vesicle Release of the BK Virus

General comments

This manuscript address the molecular and genetic aspect of BK virus. The topic addressed is interesting. I think however that there are several improvements that should be made before publication.

Specific comments

1. In the introduction, the authors would need to explain VP4.

2. In the method, the author mentioned that the cells were seeded in Cellstar 60x15mmcell culture dishes. Did the authors use non-coated dish?

3. In the result, did the authors perform immunofluorescence testing for VP1 in controls? If so, were there any differences in staining patterns between patients and controls?

4. In the result, what ethnic groups were involved in the genome integration analysis? Is there any report of easy genome integration depending on race?

5. In the discussion, the authors would better to mention the clinical strengths of this finding.

Comments on the Quality of English Language

 English language fine.

Reviewer 2 Report

Comments and Suggestions for Authors

These are very interesting research results, and we believe that the authors succeeded in establishing a BKPyV infection model in newly cultured renal tubular progenitor cell lines is particularly significant.

What is your vision for how these research results will be utilized in the future in the clinical situations?

Developing antiviral drugs or vaccines?

Identifying highly virulent virus strains??

Improving diagnostic abilities???

Reviewer 3 Report

Comments and Suggestions for Authors

I read with interest the manuscript by Daniela Gerges et al. about the description of an in vitro model to describe BK Polyomavirus (BKPyV) biology and shedding.

The study is an expansion of a previous study from the authors, and are complementary to recent studies from other labs.

The first half of the abstract (from line 15 to line 27) perfectly reflect the content of the main manuscript. However, the remaining insufficiently summarized the results or provided conclusions that are not supported by the data.

The article is mostly well written even if some details are needed to help the future readers to understand the conclusions.

Overall, the manuscript describes primary results of a new in vitro model. Unfortunately, the materials and methods miss some essential instructions to reproduce the experiments, or to fully understand the results. Moreover, most of the conclusions are not supported by the results so supplemental data, experiments and references are required.

Please see below my main suggestions to enhance the quality of the manuscript:

Major revisions:

-          In the introduction, several amalgams are made between “polyomavirus” (as Polyomaviridae family members) and the species, such as BK Polyomavirus. For example, lines 67-68 “ exocytosis appears to be regulated via various viral proteins like VP4 [16, 17]” refer to a study about SV40 and not BKPyV. Please clarify

-          Methods/cell culture and cell lines: Inclusion and exclusion criteria for study population is required.

-          Methods/Immunoreagent: Please detail the antibody production process: commercial plasmid, home-made (how?)?

-          Methods: please detail the protocol for BKPyV infection of cell lines with ultracentrifugated virions

-          Table 1: BKPyV load are highly variable according to SDs, a detailed description of the small number of individuals is needed.

-          Line 144 + lines 303-304 : mAb 4942 was not supposed to be used. Results are mentioned but not shown. Please clarify

-          Line 151-165: copy number evaluation is not detailed.

-          Line 190 + table 2 line 350: at first the authors suggest that 9 cell lines express BKPyV but in the end it seems that some infections need to be further performed.

-          Table 1: As the population is small and seem to show high variability in characteristics, the data should be expressed as Median +/- InterQuartile Range (not Mean+/-SD)

-          Lines 212-213 : “nine cell lines were successfully established, exhibiting variations in cell morphology and growth kinetics”. Details are lacking,especially since Fig1 shows only 1 cell line. Please extend in terms of division delay, passage rhythm, morphology, etc.

-          Figure 1C: Please add the results for cell lines 7 to 9.

-          Overall, the authors should use the same label for each patient/cell line/urine throughout the manuscript. A table summarizing the viral characteristics of the individuals (% decoy cells, viral loads, infectivity, …) would be very appreciated.

-          Line 232 : please provide experimental results proving that the cell cultures are pure ‘tubular cell cultures’ or rephrase throughout the manuscript.

-          Line 240-241: Please justify the scientifical soundness of the hypothesis, preferably with in silico modelization.

-          For description of confocal microscopy, the actual reported results are insufficient to support the associated conclusions. Indeed, the study only focused on VP1 or nucleus staining. All of the mentioned cellular structures (EVs, Golgi, cell membrane, …) are guessed but must be shown by immune-staing to prove the co-localization.

-          Line 302 and Figure 6: The authors assume that a truncated protein of 17 kDa weight is expressed in infected cells but the band is seen in uninfected cells, maybe with a higher signal !

-          Line 331-332 : the authors assume that “Thus, it appears that two truncated VP1 proteins (VP1a and VP1b) are produced during virion maturation, which are not included in the final BKPyV.”. To validate this conclusion, the 2 sample types would have to be run on the same immunoblot to ensure homogeneity of intensity for the 41kDa band and the presence/absence of truncated forms.

-          Line 341 + Table 2: why only five samples were analyzed ? Please discuss

-          Line 347-348: “At these sites the entire BKPyV genome could be identified”. Where are these results shown ? Which parts of the viral genome ? What are the restriction/linearization sites ? Moreover, this sentence is contradictory with the sentence line 363-365 suggesting that the authors did not find the complete-genome-sized gap.

-          Figure 8: Analyze and associated discussion of the results are required. At least, a description of adjacent human genes and of potential impact on oncogenes expression (including BKPyV Tag) is required

-          Discussion: Please discuss the use of an antibody that target genotype IV-VP1 while most of European populations are infected with genotype I and it was not evaluated on the study population

-          Line 393-394: Not all models require immortalized cell lines. Please see this reference and rephrase appropriately: AnP,Sa ́enzRobles MT,Duray AM, Cantalupo PG,Pipas JM(2019) Human polyomavirusBKVinfection ofendotheli alcells results ininterferon pathway induction and persistence.PLoS Pathog 15(1): e1007505. https://doi.org/10.1371/journal.ppat.1007505

-          Line 429-430: to support the previous conclusion, the protease cleavage might be easily explored by using protease inhibitor before immunoblotting. As they suggested, the authors should add this experiment to the manuscript.

Minor revisions:

-          Some typos must be corrected after careful review. For example “polyoma virus” line 100

-          Line 40-42 : a reference is needed

-          Lines 55-60 : the viral cycle should be better described, especially in line to understand the assemble and shedding of virions.

-          Line 67: the reference 14 is used but corresponds to an article about Parvovirus (?)

-          Line 71 uses reference 20 to justify the role of EVs to help immune evasion in BKPyV infection. However, this reference is about the JCPyV whereas reference 28 refuted such property for BKPyV.

-          Lines 189-190: Description of the unestablished cell cultures would be useful to understand why the protocol is not effective for all of the patients.

-          Line 111 : the specificity of mAb 4942 must be defined (specific for a genotype ? for BKPyV ? for VP1 ?) and justified by relevant references

-          Line 117: described above ?

-          Line 122-123: how can such a difference in dilution between the 2 antibodies be justified? initial concentration? If so, please describe the protocol with concentrations and not dilutions, especially since the manuscript aims to show distinct staining profiles between the 2 antibodies.

-          Line 194 : “(3.275)” what does this correspond to ?

-          Line 199 + table 1 : please express and normalize the viral loads as “copies per mL”

-          Line 202-203: please explain the principle and why an immune-histochemistry for SV40, and not for BKPyV, was performed, particularly since only one biopsy was positive.

-          Figure 1B: Since there are only 9 cell lines that are described as “exhibiting variations in cell morphology and growth kinetics”, the results should be entirely shown, not only a “representative” cell line.

-          Figure 2-3-4-5: please annotate the figure in accordance with the legend (rows and columns).

-          Figure 3-4-5: please annotate the figure in accordance with the legend to highlight the elements looked for

-          Line 252-253: this sentence does not sound relevant in the results. This should be moved to the Discussion, or suppressed if not useful for interpretation of the results.

-          Lines 292-293 : “Visualizing VP1 with both a monoclonal and a polyclonal antibody revealed different staining patterns during virus maturation” must be rephrased. The assumption made in the previous paragraph cannot be taken as true immediately afterwards. All the more so as the previous results concern a one-off observation and not a real assessment of viral maturation over time.

-          Line 314: JC Polyomavirus or JCPyV, not “JC virus”

-          Line 327:

o   what is this negative control? The same as in figure 6?

o   Left wing results should be presented at first

o   Why using a label A/B/C while numbers are used in the other parts of the manuscript? Please refer to the major comments to homogenize the labelling of individuals.

-          Figure 8 legend: please clarify which passage was used to describe the results: 3 or 4 ?

Round 2

Reviewer 3 Report

Comments and Suggestions for Authors

I read with attention this extensively revised manuscript. Many theorical and experimental data have been added to enhance the overall quality and scientifical soundness of the article.

All of the reviewers' comments have been taken into account.

Personally, my minor comments have all been implemented. Most of the major comments have also been implemented. However, a few more minor corrections are still needed. Here are my responses, point by point:

(C1b) The authors have corrected the abstract appropriately

(C2b) Even if SV40 and BKPyV show biological similarities, this could not be extended to all Polyomavirus. In fact, biology of Merkel Cell Polyomavirus is completely different and recent findings imply distinct features of JCPyV cellular signaling. Thus, I maintain my request to clarify the introduction by citing the relative Polyomavirus for each reference, if this does not concern the BKPyV, which is the studied species in the present manuscript.

(C3b) The number of individuals included and excluded would also be relevant information.

(C4b) The authors provided the required information.

(C5b) The authors provided the required information.

(C6b) This  table is highly relevant. The authors must precise that the BKPyV load corresponds to copies per mL

(C7b) The authors corrected the text adequately.

(C8b) The authors provided relevant method details.

(C9b) The authors corrected the text adequately.

(C10b) The authors corrected the data adequately. The authors must precise that the BKPyV load corresponds to copies per mL

(C11b) The authors corrected the data adequately.

(C12b) The authors corrected the data adequately.

(C13b) The authors corrected the data adequately.

(C10b) The authors revised this part adequately and provided robust data to justify their hypothesis. However, please explain the conflicting results of protein sizes between immunoblots and modelization (3.4 : 17 kDa and 36-37 kDA; 3.5 : 18 kDa and 35 kDa). Moreover, in part 3.4, VP1a is described as the smallest protein whereas this is the biggest protein in part 3.5. Please correct.

(C15b) The authors provided relevant experimental results to support their conclusions.

(C16b) Thanks to the authors for the reply but this does not explain why a band was found in uninfected cells. Removing this surprising result is not relevant.

(C17b) The authors provided relevant experimental results to support their conclusions.

(C18b) The authors corrected the manuscript adequately.

(C18b) The authors corrected the manuscript adequately.

(C19b) The authors corrected the manuscript adequately.

(C20b) The authors corrected the manuscript adequately.

(C21b) The authors provided relevant data to justify their statement.

(C22b) The authors corrected the manuscript adequately.

(C23b) The authors corrected the manuscript adequately.
